# Medication errors during simulated paediatric resuscitations: a prospective, observational human reliability analysis

Nicholas Appelbaum [1,2,3] Jonathan Clarke [2,3,4] Calandra Feather [1,2,3] Bryony Franklin,[2,5] Ruchi Sinha,[6] Phillip Pratt,[3] Ian Maconochie,[7] Ara Darzi[1,2,3]

For numbered affiliations see end of article.

**Correspondence to**
Dr Nicholas Appelbaum;
n.appelbaum@imperial.ac.uk

### ABSTRACT

**Introduction** Medication errors during paediatric resuscitation are thought to be common. However, there is little evidence about the individual process steps that contribute to such medication errors in this context.

**Objectives** To describe the incidence, nature and severity of medication errors in simulated paediatric resuscitations, and to employ human reliability analysis to understand the contribution of discrepancies in individual process steps to the occurrence of these errors.

**Methods** We conducted a prospective observational study of simulated resuscitations subjected to video microanalysis, identification of medication errors, severity assessment and human reliability analysis in a large English teaching hospital. Fifteen resuscitation teams of two doctors and two nurses each conducted one of two simulated paediatric resuscitation scenarios.

**Results** At least one medication error was observed in every simulated case, and a large magnitude (>25% discrepant) or clinically significant error in 11 of 15 cases. Medication errors were observed in 29% of 180 simulated medication administrations, 40% of which considered to be moderate or severe. These errors were the result of 884 observed discrepancies at a number of steps in the drug ordering, preparation and administration stages of medication use, 8% of which made a major contribution to a resultant medication error. Most errors were introduced by discrepancies during drug preparation and administration.

**Conclusions** Medication errors were common with a considerable proportion likely to result in patient harm. There is an urgent need to optimise existing systems and to commission research into new approaches to increase the reliability of human interactions during administration of medication in the paediatric emergency setting.

### INTRODUCTION
#### Background
Medication errors are among the leading cause of avoidable harm in healthcare worldwide[1] and up to three times more common in children than in adults.[2] The paediatric emergency environment, characterised by urgency and fraught with interruptions, is one of the clinical areas most vulnerable to error. Medication administration in emergencies is complex as it requires successful

### Strengths and limitations of this study

► This study adds to the literature base, highlighting paediatric emergency medication error as worrying, potentially highly harmful and requiring urgent attention.

► This study is one of the first to use Human Reliability Analysis to link task discrepancies with resultant medication errors, as well as to link these discrepancies directly to potential harm. This effort has demonstrated that a significant fraction of the burden of error in the paediatric emergency drug administration process originates during the preparation and administration phase and that most of these errors are likely to be undetected in clinical practice.

► Although we went to considerable lengths to replicate the paediatric emergency environment, the simulation environment cannot truly reflect the clinical environment during a genuine emergency.

► Furthermore, this study was conducted at a single site and participants were not blinded to the purpose of the study, so it is potentially subject to preparation bias.

► Participants were recruited from the paediatric emergency unit, intensive care unit and general paediatrics ward and had variable experience of emergency cases. However, all participants worked in clinical units that manage critically ill children.

interactions between different teams of medical and nursing staff, as well as between individual members of these teams. An additional challenge relates to obtaining relevant medication information and translating this into the required dose and concentration of the correct drug to be administered by the correct route for the clinical indication, all in a necessarily short space of time. Medication errors in general, and medication administration errors in particular, are both under-detected and under-reported,[3] such that little is known of their incidence or impact during resuscitation. However, medication errors have been reported in 7 out of 10 simulated paediatric resuscitations,[4] with other recent

simulation studies suggesting 26%[5] to 70%[6] of administered medicines being given at the wrong dose. Laboratory studies analysing syringes prepared for anaesthetic use have found at least 15% to be greater than 20% discrepant from the intended drug concentration.[7]

The broader, systems view of medical error, heralded by the Institute of Medicine's 'To Err is Human' report, saw the widespread adoption of Reason's organisational accident model[8] in healthcare. More recently, human reliability analysis (HRA) techniques, previously commonplace only in other high risk industries, have become increasingly used in healthcare research.[9 10] HRA is based on the understanding that neither humans nor systems can be error-proof, and asserts that to improve safety and reliability, a thorough analysis of system vulnerabilities at a task level is needed, taking into account human-human and human-machine interactions.[11 12] Medication safety researchers have previously used an HRA technique, the systematic human error reduction and prediction approach (SHERPA), to identify system vulnerabilities in ward-based medication administration,[13] anaesthesia[14] and general surgery.[15] This approach, however, has not been used quantitatively in medication safety research and has not been applied to paediatric resuscitation.

Our objectives were to describe the incidence, nature and severity of medication errors in simulated paediatric resuscitations, and then use HRA to understand the contributory role played by individual process step discrepancies with a focus on those discrepancies contributing to large magnitude and/or clinically significant errors.

## METHODS

### Study design and setting

This prospective observational study was conducted from April 2017 to November 2017 in a medical simulation facility within a large teaching hospital. The hospital has a paediatric emergency department (seeing 27 000 children each year) and a comprehensive paediatric inpatient service (admitting 5000 each year). The hospital used electronic prescribing in the inpatient setting, but during resuscitations, medications were more commonly ordered on paper prescription charts. We recruited resuscitation teams of four clinicians, that were randomised to participate in one of two standardised simulated paediatric resuscitation scenarios. The study was approved by the Health Research Authority and the hospital concerned. National Health Service ethics approval was not required. Participants gave written informed consent.

### Patient and public involvement

The research team held a workshop with parents to get their feedback on the proposal, develop the patient and public involvement and engagement (PPIE) plans, and identify future areas for research. We actively sought attendees through INVOLVE's 'People in Research' website, social media and Imperial College London's existing networks. Our team has also participated in a PPIE event run by the Royal College of Paediatrics and Child Health, in collaboration with MedsIQ (http://www.medsiq.org) and Medicines for Children (http://www.medicinesforchildren.org.uk), two UK-based paediatric medication safety initiatives.

### Participants

Eligible participants were a convenience sample of medical and nursing staff from the departments of paediatrics and paediatric emergency medicine at the study hospital. Participants were assigned into teams comprising a senior doctor (a specialist registrar, with at least a year of prior experience as a registrar), a junior doctor, a senior nurse (with at least 5 years' nursing experience) and a junior nurse.

### Clinical scenarios

The two scenarios were:
1. Prolonged status epilepticus in an 8-month-old, 8 kg child.
2. Presumed meningococcal sepsis in a 10-month-old, 9 kg child.

The two scenarios (online supplementary appendix 1) were designed by a collaboration of paediatric nurses, emergency physicians, intensivists, general paediatricians and anaesthetists. Face validity was established by an independent expert panel of six, with representation from each of these professional groups, including two lead paediatric clinical nurse educators. It was deemed that the two scenarios were both similarly demanding and clinically sound, with treatment recommendations corresponding closely to UK Resuscitation Council and Royal College of Paediatrics and Child Health teaching cases.

A simulated paediatric resuscitation bay was created. The mannequin used was a SimBaby V.2 (Laerdal Medical, Stavanger, Norway), and the syringe pump stack consisted of Alaris PK MK4 units (Becton Dickinson, Franklin Lakes, USA). All relevant print materials (eg, British National Formulary for Children[16] and local guidelines) and hospital information technology systems as well as external internet access were available. Participants were requested to prescribe, prepare and administer medications exactly as per usual practice, to use mobile applications or websites as they would in clinical practice and to telephone specialist colleagues if required. A paediatric intensivist, the hospital lead for paediatric simulation, ran the scenarios. She provided standardised clinical information as live feedback and answered questions regarding the child's response to treatment or their current condition when needed.

### Data sources and measurement

A Scotia Medical Observation Training System (smots, Scotia UK, Edinburgh, UK), with two 3-axis, ceiling-mounted video cameras, and three mobile, high-definition cameras equipped with boom microphones, was used. Both nurses in each team wore head-mounted

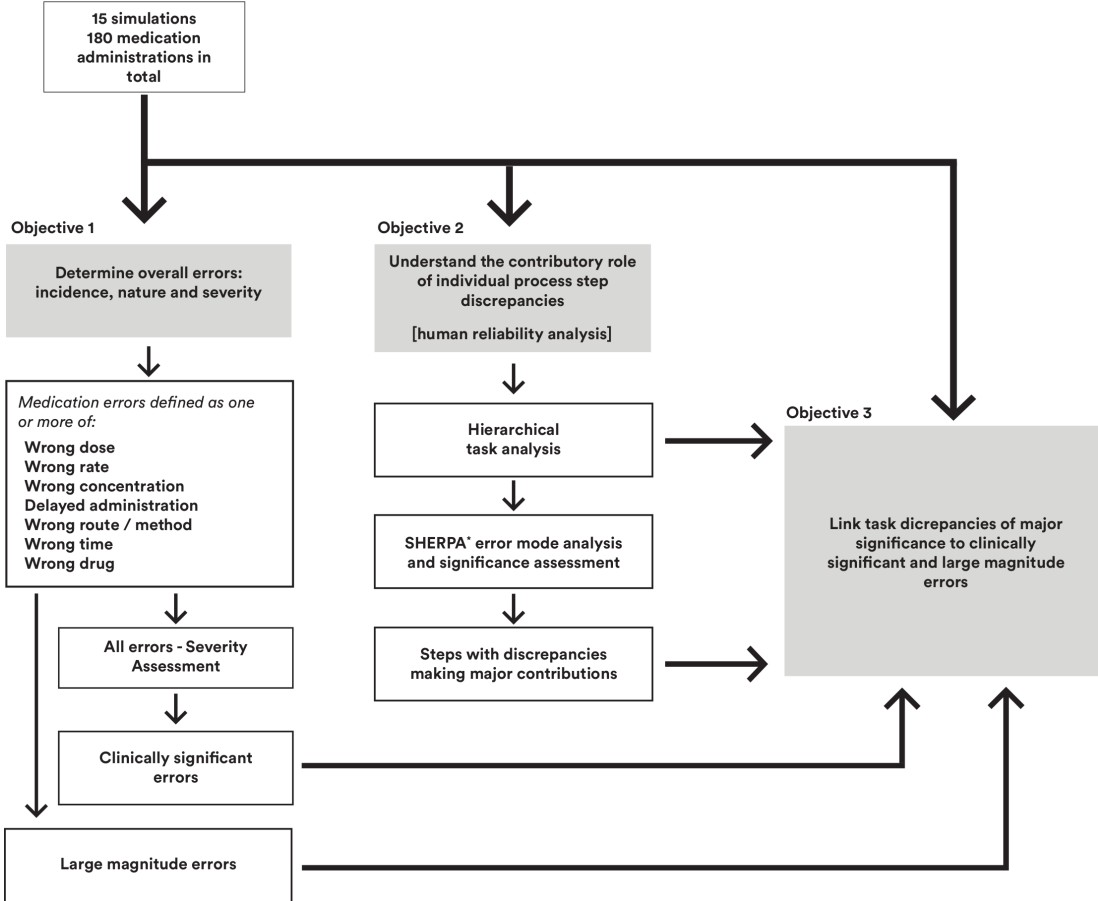

**Figure 1** Overview of study objectives and analyses. *SHERPA, systematic human error reduction and prediction approach.

high-definition video cameras (GoPro Inc, California, USA). The video recordings were analysed by a research nurse with 10 years' experience in paediatric intensive care.

### Outcome measures

We reserved the term 'medication error' to describe an overall error with respect to a particular drug administration as a whole, after having been administered to a patient, and the term 'discrepancy' to refer to observed deviations from expected local practice at the level of the individual task. This approach has been successfully used by other medication safety researchers.[17]

Variation in clinician practice and human adaption to the complex process of medication ordering, preparation and administration typically results in many minor task discrepancies that may not individually, or even in combination, result in a medication error or patient harm.[17] Other discrepancies almost always result in medication errors. For example, a prescribing or pump-programming discrepancy is highly likely to result in a medication error. To identify the most important task discrepancies, we assessed all observed discrepancies to establish the extent to which they may or may not have contributed to any resultant medication errors.

Figure 1 summarises the study objectives and associated analyses.

### Medication errors

Medication errors included any errors in dose, administration rate, concentration, drug, route of administration, method of administration, timing or delay in administration. Operational definitions for each of these are given in online supplementary appendix 2. Briefly, dosing errors were defined as a greater than 10% deviation from the recommended dosing range (DRDR)[18] at the study site. Any deviation from the recommended rate of administration (DRDRate) was calculated in a similar manner and deviations of more than 10% were considered to be medication errors. Where there was a greater than 25% discrepancy in the DRDR or DRDRate, the errors were considered as 'large magnitude'. Deviations from the recommended concentration (DRC) of greater than 10% from the concentration specified in local guidance were also included as medication errors.

To identify delayed administrations, the time taken for the dose to be 'ready for delivery' was calculated as the time for the doctors to obtain any medication information required plus the nurse-led preparation time. The time to be ready for deliver was considered 'prolonged' when a particular team took more than double the median time for that specific drug across the entire study without clinical cause for the delay as determined by the nurse assessor. For example, if a medication administration was

interrupted to reassess the patient clinically or to administer another medication as a priority, a prolonged time would be excluded as an error on clinical grounds.

### Severity assessment

There are few validated tools that can be used to assess the potential severity of medication errors without knowledge of patient outcomes and that are thus usable in simulated studies.[19] One of these tools is that of Dean and Barber, based on four to five experts independently assessing each error on a 0 to 10 scale, and their mean score used as an index of severity. Mean scores under 3 suggest errors of minor severity, those between 3 and 7 as moderate and those greater than 7 as severe.[20] We used this approach, with two paediatric intensivists, one paediatric anaesthetist, one senior critical care nurse and one senior clinical pharmacist assessing each error.

### Discrepancies at the level of the task

A hierarchical task analysis (HTA) was developed based on a similar framework for ward-based medication administration[13] and assessed for face validity by five senior nurses in the study hospital. A generic human error taxonomy, based on the SHERPA external error modes[21] with one additional error mode, 'information not sought', was used to code observed discrepancies against the HTA.

Where there were more than two discrepancies at a single step for a specific administration, the nurse assessor made a subjective assessment of which had the greater overall consequence and assigned an error mode (figure 2) to that discrepancy only. To capture 'root-cause' system vulnerabilities, steps where an action was performed correctly, but which perpetuated a previous medication error, were not classed as discrepancies. An example would be a correct volume calculation based on an incorrectly prescribed dose. In this example, the volume calculation which persisted but did not directly cause the error wouldn't be classed as a discrepancy whereas the incorrect prescription would be.

### Significance assessment of task discrepancies

All task discrepancies were classified by the nurse assessor according to the contribution made by the discrepancy as follows:

**No contribution:** the discrepancy did not contribute to a medication error.

**Minor contribution:** some contribution made to a medication error.

**Major contribution:** the task discrepancy led directly to a medication error.

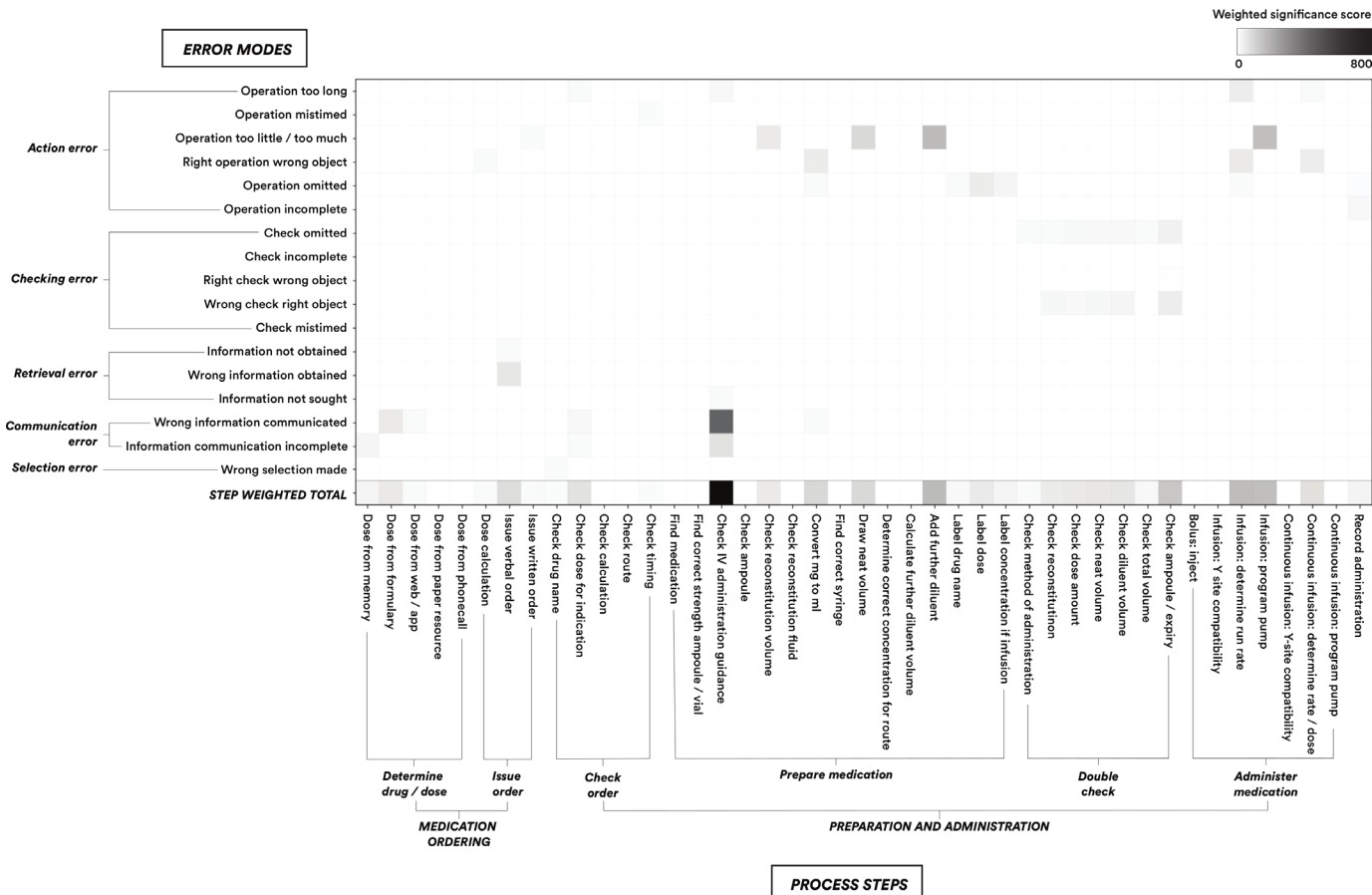

**Figure 2** Significance-weighted heat-map of error by process step and SHERPA error mode. IV, intravenous; SHERPA, systematic human error reduction and prediction approach.

## Data management and analysis

Medication errors were described according to the type of error, method of administration (eg, injection, infusion, continuous infusion) and stage of the medication use process (eg, medication ordering, preparation, administration) during which the error occurred. Error rates were calculated using the total number of applicable administrations as the denominator.

Step discrepancies were presented as counts, grouped by task and contribution to medication error. Discrepancy rates were calculated as the percentage of discrepancies that made a major, minor or no contribution to an error, with the number of observed discrepancies at each process step as the denominator. Of those discrepancies making major contributions to medication errors, the proportion that led to clinically significant errors (severity score >3) and/or large magnitude errors (DRDR or DRDRate >25%) was also calculated.

There is no literature that quantifies the extent to which a step discrepancy having a minor or major contribution to an error is of greater significance than a step discrepancy that makes no contribution to an error. For a weighted, 'heat map' HRA analysis, it was therefore necessary to attribute different weights to discrepancies that resulted in error to those that did not. Substep discrepancies were therefore weighted, agreed by the expert panel, as follows:

► No contribution: weight=1
► Minor contribution: weight=10.
► Major contribution: weight=40.

The total weighted significance score for each step was thereby calculated for each error mode.

### Interobserver reliability

One of the 15 simulations was reanalysed by an additional independent nurse not involved in the simulations. Spearman's rank correlation coefficient was calculated for 38 continuous variables (eg, doses, elapsed times) and Cohen's kappa for 20 categorical variables (eg, reconstitution fluid, parameters for labelling quality assessment). The data set concerning the medication errors and timing parameters was treated separately from the data set containing the parameters for the HRA (the task discrepancies).

## RESULTS

Data were collected during 15 simulations according to participant availability. Eight simulations were of the prolonged seizures case, and seven of the meningococcal sepsis case. Participants comprised 30 doctors and 30 nurses (table 1), each of whom completed one simulation. For categorical variables, Cohen's kappa values were between 0.862 (medication error) and 0.954 (HRA). Continuous variables were only present in the medication error data set, for which Spearman's rank coefficient was greater than 0.904 for all variables.

**Table 1** Characteristics of study population

|  | Overall | Doctors | Nurses |
|---|---|---|---|
| Total number of participants | 60 | 30 | 30 |
| Age * |  |  |  |
| Median (range) | 30 (23–51) | 30.5 (23–44) | 28 (23–51) |
| Gender (%) |  |  |  |
| Female | 52 (87%) | 23 (77%) | 29 (97%) |
| Male | 8 (13%) | 7 (23%) | 1 (3%) |
| Years in clinical practice (n, %) |  |  |  |
| 0–5 | 32 (53%) | 11 (37%) | 21 (70%) |
| 6–10 | 20 (33%) | 16 (53%) | 4 (13%) |
| 11–15 | 5 (8%) | 3 (10%) | 2 (7%) |
| 16–20 | 1 (2%) | 0 (0%) | 1 (3%) |
| >20 | 2 (3%) | 0 (0%) | 2 (7%) |
| Years in paediatric practice (n, %) |  |  |  |
| 0–5 | 37 (62%) | 16 (53%) | 21 (70%) |
| 6–10 | 16 (27%) | 12 (40%) | 4 (13%) |
| 11–15 | 5 (8%) | 2 (7%) | 3 (10%) |
| 16–20 | 1 (2%) | 0 (0%) | 1 (3%) |
| >20 | 1 (2%) | 0 (0%) | 1 (3%) |

*Age data was omitted for five participants.

## Medication errors

Participants conducted 180 medication administrations. Overall, errors were observed reaching the patient for 52 drug administrations (29%) and at least once in every simulation. Of these errors, 30 (58%) were assessed as being of minor severity, 16 (31%) as moderate and 6 (12%) as severe. There were 27 large magnitude errors (52% of all errors), in which the DRDR/DRDRate was greater than 25%. Of all erroneous administrations that reached the patient, only two (4%) were noticed by staff after administration and therefore may have been reported in clinical practice. A detailed error analysis is provided in table 2 and description of the 10 most severe errors in table 3.

## Hierarchical task analysis

The full HTA is shown as figure 3 and shows all steps assessed in the paediatric emergency drug administration process.

## Errors and discrepancies by stage of medication use and process substep

Overall, 884 step discrepancies were observed, excluding dependent downstream discrepancies after an initial discrepancy. Of these 884 step discrepancies, 174 (20%) were linked to a medication error, with 70 (8%) assessed as making a major contribution to an error, 104 (12%) making a minor contribution and 710 (80%) making no contribution.

**Table 2** Incidence, nature and severity of errors, presented by phase and type of error and method of administration

| | Number of errors | Incidence as % of total administrations (n=180) | Incidence as % of administrations by method of administration | Severity assessment | | | |
| --- | --- | --- | --- | --- | --- | --- | --- |
| | | | | Minor (severity score <3) (n) | Moderate (severity score 3–7) (n) | Severe (severity score >7) (n) | Mean severity score |
| **Any error** | **52** | **29%** | | **30** | **16** | **6** | **3.2** |
| By stage of medication use and error type | | | | | | | |
| Prescription errors | 8 | 4% | | 2 | 4 | 2 | 5.1 |
| Wrong dose | 8 | 4% | – | 2 | 4 | 2 | 5.1 |
| Wrong route | 1 | 1% | – | – | – | 1 | – |
| Wrong drug | – | – | – | – | – | – | – |
| Preparation and administration errors | 40 | 22% | | 25 | 12 | 5 | 3.1 |
| Wrong drug | 1 | 1% | – | – | – | 1 | – |
| Wrong dose prepared | 10 | 6% | – | 6 | 3 | 1 | 2.8 |
| Wrong diluent/concentration | 20 | 11% | – | 12 | 6 | 2 | 3.0 |
| Wrong rate (infusions) | 11 | 6% | – | 4 | 5 | 2 | 4.2 |
| Wrong route/method | 9 | 5% | – | 5 | 3 | 1 | 3.6 |
| Wrong time | 1 | 1% | – | – | – | 1 | – |
| By method of administration and error magnitude | | | | | | | |
| Bolus doses (n=77 in total) | | | | | | | |
| Any error | 24 | – | 31% | 13 | 7 | 4 | 3.6 |
| Dose error, DRDR*>10% | 7 | – | 9% | – | 5 | 2 | 5.0 |
| Dose error, DRDR*>25% | 6 | – | 8% | – | 4 | 2 | 5.7 |
| Delayed administration, dose correct | 5 | – | 7% | 2 | 2 | 1 | 4.2 |
| Delayed administration, dose error | – | – | – | – | – | – | – |
| Intermittent infusions (n=48 in total) | | | | | | | |
| Any error | 18 | – | 38% | 10 | 7 | 1 | 3.2 |
| Total dose error, DRDR*>10% | 5 | – | 10% | 3 | 2 | – | 3.7 |
| Total dose error, DRDR*>25% | 3 | – | 6% | 1 | 2 | – | 5.2 |
| Rate error, DRDRate‡>10%† | 8 | – | 17% | 2 | 5 | 1 | 4.6 |
| Rate error, DRDRate‡>25%† | 6 | – | 13% | – | 5 | 1 | 6.0 |
| Delayed administration, correct dose and rate | 2 | – | 4% | 2 | – | – | 1.4 |
| Delayed administration, incorrect dose or rate | 1 | – | 2% | – | 1 | – | – |
| Continuous infusions (n=55 in total) | | | | | | | |
| Any error | 10 | – | 18% | 7 | 2 | 1 | 2.2 |
| Delivery rate error, DRDRate‡>10% | 5 | – | 9% | 2 | 2 | 1 | 3.4 |
| Delivery rate error, DRDRate‡>50% | 5 | – | 9% | 2 | 2 | 1 | 3.4 |
| Delayed administration, correct delivery rate | 1 | – | 2% | 1 | – | – | – |
| Delayed administration, with incorrect delivery rate | 1 | – | 2% | – | 1 | – | – |

More than one error type can occur in one medication administration and a single medication error may meet more than one criterion, so individual error types do not sum to the total by stage or method. Other error types (for example diluent errors) are included in the 'any error' counts but are not presented as sub-counts.

*DRDR (deviation from recommended dosing range)=absolute value of the percentage difference from the recommended dose or dose range.
†Rate errors are only shown for intermittent infusions where the delivery rate error is due to a pump-programming error.
‡DRDRate (deviation from recommended dosing rate)=absolute value of the percentage difference from the recommended rate of administration.

Figure 2 shows the significance-weighted HRA data represented as a heat-map demonstrating the relative contributions of discrepancies at each step and by each error mode to medication errors. Table 4 summarises the discrepancy counts per step as well as the percentage of both large magnitude and clinically

**Table 3** Details of the 10 most severe medication errors by severity score

| Rank | Medication | Stage | DRDR* | DRDRate† | Other error | Severity score | Error detail | Error cause |
|---|---|---|---|---|---|---|---|---|
| 1 | Thiopentone | Ordering | 900% | – | – | 8.8 | 320 mg rather than 32 mg given as IV bolus | Doctors asked for two doses of thiopentone to be prepared which were ordered at 32 mg each. Nurse read phenytoin dose, which was 160 mg, written on the line above the thiopentone order on the medication chart. Two boluses of 160 mg rather than 32 mg were prepared and administered. |
| 2 | Calcium chloride | Ordering | 789% | – | – | 8.6 | 8.8 mmol rather than 0.99 mmol given as IV bolus | Incorrect dose for indication selected and prescribed from British National Formulary (BNF) |
| 3 | Thiopentone | Administration | – | – | Timing error | 8.2 | Given prior to anaesthetist ready | Medication prepared and administered prior to medical and anaesthetic team being ready to manage airway and breathing |
| 4 | Dextrose | Preparation | – | – | 14 min to administer | 7.5 | 825% of median time for dextrose. Hypoglycaemic patient, glucose=2.2 | Excessive time spent working out required volume to administer due to confusion caused when checking concentration information across multiple resources |
| 5 | Epinephrine | Preparation | – | –88.9% | 18 min taken to administer | 7.5 | 286% of median time for epinephrine infusions | Team unfamiliar with prescribing and preparing epinephrine infusion, time spent accessing multiple resources |
| 6 | Phenytoin | Preparation/ administration | – | 300% | – | 7.2 | Correct dose given undiluted at four times the recommended administration rate (4 mg/kg/min rather than 1 mg/kg/min) | Calculation error when setting up rate on pump, administered over 5 min instead of 20 min. Undiluted phenytoin infusions are not in accordance with local policy but this was not clearly indicated in the IV administration guidance |
| 7 | Phenytoin | Administration | – | –99.2% | – | 6.9 | Correct dose given too slowly (0.0078 mg/kg/min rather than 1 mg/kg/min) | Calculation error when setting up rate on pump |
| 8 | Aciclovir | Ordering and administraiton | –45.5% | 163.3% | – | 6.8 | 105 mg rather than 210 mg given as infusion and run over 20 mins instead of an hour | Incorrect dose chosen for indication from BNF and administered over 20 mins instead of 1 hour. Correct dose was on following page of BNF |
| 9 | Aciclovir | Ordering | –45.5% | – | – | 6.2 | 105 mg instead of 210 mg | Incorrect dose selected for indication from BNF. Correct dose was on following page of BNF |
| 10 | Aciclovir | Preparation/ administration | – | high | – | 5.7 | Given as a neat bolus rather than being diluted and infused | Administered as bolus instead of as an infusion over 1 hour |

*DRDR (deviation from recommended dose or dose range)=percentage difference from the recommended dose or dose range.
†DRDRate (deviation from recommended dosing rate)=percentage difference from the recommended rate of administration.
IV, intravenous.

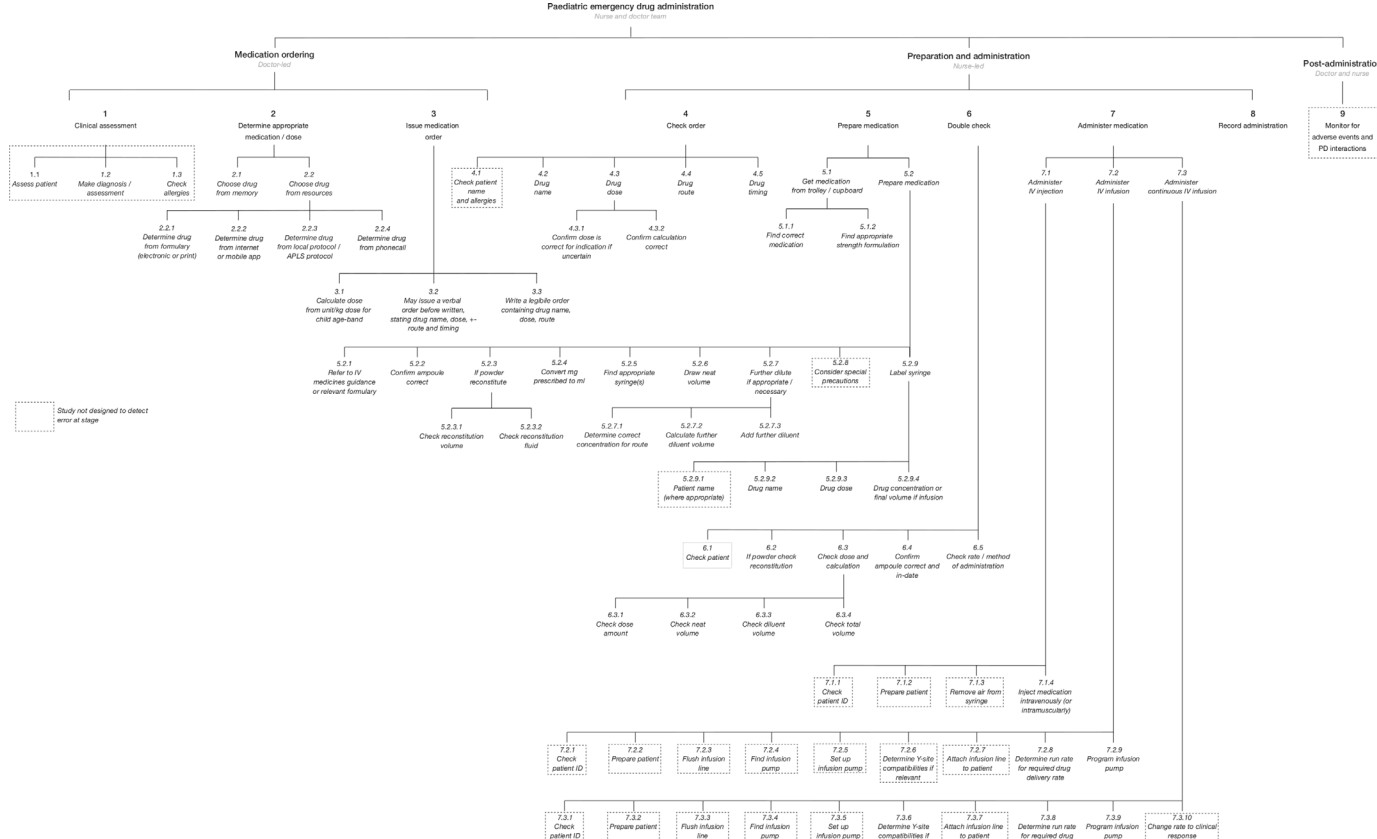

**Figure 3** The hierarchical task analysis.

significant errors with a major contribution made at each HTA step.

### Errors and discrepancies during medication ordering

We observed 170 discrepancies during the ordering phase. Five of the 22 clinically significant medication errors were due to discrepancies during medication ordering, with three of these due to incorrect dose information retrieval from the British National Formulary for Children. The majority of the remaining discrepancies (136) were due to incomplete verbal medication orders based on which drug preparation commenced, two of which resulted in medication errors. Of the 180 written medication orders examined, there were six discrepancies observed, all of which were corrected by the clinicians and therefore made no contribution to any dosing errors, but did result in one delayed administration.

### Errors and discrepancies during medication preparation

During medication preparation, 310 discrepancies were observed, representing 35% of all observed discrepancies. These contributed to one medication error involving the wrong drug, 10 dose errors and 20 diluent or dilution errors. The retrieval of preparation and administration information from the online intravenous medications guide was the step mostly likely to contribute to medication errors during medication preparation, with 42 discrepancies (19 major contribution to a medication error, one minor), resulting in nine clinically significant

medication errors (severity score >3). The retrieval of incorrect information and taking an excessively long time to identify the correct information within the guidance were the most common discrepancy error modes.

There were seven discrepancies when converting milligrams to millilitres of undiluted drug, and 18 discrepancies (six making a major contribution) when, after having made the correct calculations, nurses withdrew either the incorrect amount of undiluted drug or the incorrect amount of diluent into the syringe.

### Double-checking

Overall, 259 discrepancies were observed during the double-checking phase, 72 (28%) of which made a minor contribution to a medication error. Checking the route and method (eg, infusion or bolus) of administration was the most frequently omitted.

We observed 29 medication errors that were made during medication ordering and preparation but had not been yet administered to the patient at the point of double-checking. These errors were thus potentially 'interceptable' but all ultimately reached the patient. Of these errors, in 14 cases, the double-checking interaction between the nurses included the incorrect step but failed to identify it as incorrect.

### Errors and discrepancies during the administration phase

Of all observed discrepancies, only 28 (3%) occurred during administration. These resulted in 11 wrong rate

**Table 4** Number, frequency and relationship of discrepancies to resultant errors with subanalysis of major discrepancies that resulted in large magnitude and clinically significant errors

| Stage of deviation | Total discrepancies (n) | Relationship to resultant errors | | Discrepancies which made a major contribution to a medication error | | |
|---|---|---|---|---|---|---|
| | | No contribution n (%*) | Minor contribution n (%*) | Major contribution, total (n, %*) | Discrepancies that resulted in clinically significant errors (n, %*) † | Discrepancies that resulted in large magnitude errors n (%*) ‡ |
| Overall | 884 | 710 (80) | 104 (12) | 70 (8) | 33 (47) | 31 (44) |
| Ordering phase | 170 | 159 (94) | 2 (1) | 9 (5) | 5 (23) | 6 (22) |
| Determine dose | 29 | 21 (72) | 1 (3) | 7 (24) | 5 (23) | 6 (22) |
| Dose from memory | 4 | 2 (50) | – | 2 (50) | – | 1 (4) |
| Dose from formulary | 10 | 6 (60) | 1 (10) | 3 (30) | 3 (14) | 3 (11) |
| Dose from other resource | 11 | 10 (91) | – | 1 (9) | 1 (5) | 1 (4) |
| Dose calculation | 4 | 3 (75) | – | 1 (25) | 1 (5) | 1 (4) |
| Issue order | 141 | 138 (98) | 1 (1) | 2 (1) | – | – |
| Issue verbal order | 136 | 134 (98) | 1 (1) | 1 (1) | – | – |
| Issue written order | 5 | 4 (80) | – | 1 (20) | – | – |
| Preparation phase | 588 | 456 (78) | 89 (15) | 43 (7) | 17 (77) | 13 (48) |
| Check order | 19 | 6 (32) | 9 (47) | 4 (21) | 3 (14) | 1 (4) |
| Check drug name | 1 | – | – | 1 (100) | 1 (5) | 1 (4) |
| Check dose for indication | 15 | 5 (33) | 8 (53) | 2 (13) | 1 (5) | – |
| Check calculation | 1 | – | 1 (100) | – | – | – |
| Check route and timing | 2 | 1 (50) | – | 1 (50) | 1 (5) | – |
| Preparation, actual | 310 | 263 (85) | 8 (3) | 39 (13) | 14 (64) | 12 (44) |
| Find correct medication and strength of vial | 2 | – | 2 (100) | – | – | – |
| Check intravenous administration guidance | 42 | 22 (52) | 1 (2) | 19 (45) | 9 (41) | 6 (22) |
| Check ampoule | 1 | – | 1 (100) | – | – | – |
| Check reconstitution fluid and volume | 7 | 3 (43) | 1 (14) | 3 (43) | – | – |
| Convert milligrams to millilitres | 7 | 2 (29) | – | 5 (71) | 3 (14) | 4 (15) |
| Find correct syringe and draw neat volume | 8 | 3 (38) | – | 5 (63) | 1 (5) | 1 (4) |
| Determine correct concentration for route | 2 | 2 (100) | – | – | – | – |
| Calculate further diluent volume and dilute | 12 | 4 (33) | 1 (8) | 7 (58) | 1 (5) | 1 (4) |
| Label syringe with drug name, dose and concentration if appropriate | 231 | 227 (98) | 4 (2) | – | – | – |
| Double checking | 259 | 187 (72) | 72 (27) | – | – | – |
| Check reconstitution | 9 | 5 (57) | 4 (44) | – | – | – |
| Check dose and neat volume | 69 | 48 (70) | 21 (30) | – | – | – |
| Check diluent volume and total volume separately | 86 | 63 (73) | 23 (27) | – | – | – |
| Check ampoule+expiry | 29 | 26 (90) | 3 (10) | – | – | – |
| Check rate/method | 66 | 45 (68) | 21 (32) | – | – | – |

Continued

**Table 4** Continued

| Stage of deviation | Total discrepancies (n) | **Relationship to resultant errors** | | **Discrepancies which made a major contribution to a medication error** | | |
|---|---|---|---|---|---|---|
| | | No contribution n (%*) | Minor contribution n (%*) | Major contribution, total (n, %*) | Discrepancies that resulted in clinically significant errors (n, %*) † | Discrepancies that resulted in large magnitude errors n (%*) ‡ |
| Administration phase | 28 | 8 (29) | 2 (7) | 18 (64) | 11 (50) | 12 (44) |
| Boluses: inject | 2 | 1 (50) | 1 (50) | – | – | – |
| Infusions: determine run rate | 8 | – | 1 (13) | 7 (88) | 5 (23) | 5 (19) |
| Determine Y-site compatibility | 2 | 2 (100) | – | – | – | – |
| Determine delivery rate for continuous infusions | 7 | 3 (43) | – | 4 (57) | 3 (14) | 4 (15) |
| Programme infusion pump | 9 | 2 (22) | – | 7 (78) | 3 (14) | 3 (11) |
| Recording | 88 | 88 (100) | – | – | – | – |

*The denominator for percentages is the total number of discrepancies in each row.
†Number and percentage of clinically significant errors (severity score >3) with major contributory discrepancies made at each specific step, of a total of 22 clinically significant errors.
‡Number and percentage of large magnitude errors (deviation from recommended dosing range (DRDR) or deviation from recommended dosing rate (DRDRate) >25%) with major contributory discrepancies made at each specific step, of a total of 27 large magnitude errors.

errors (five moderate, two severe), nine wrong method errors (three moderate, one severe) and one severe wrong time error. Discrepancies during the administration phase constituted a third of all discrepancies that made a major contribution to a clinically significant error.

Infusions in particular were prone to administration errors. Of the 17 discrepancies observed during infusion rate calculations or when programming the infusion pump for intermittent infusions, 14 were of major consequence, and accounted for 23% of all clinically severe errors. Seven discrepancies (four making a major contribution to a medication error) occurred when determining the delivery rate for continuous infusions.

## DISCUSSION

This prospective observational study is the first in paediatric emergency medicine to include a quantitative HRA, allowing identification of the task discrepancies making the greatest contribution to medication error. We identified at least one medication error in all 15 simulations, and a large magnitude or clinically significant error in 12 of these.

### Comparison with previous literature

Historical heterogeneity of the definitions of medication error and the variability in reporting metrics make comparison with previous literature difficult.[22 23] Additionally, there are few simulated studies and no relevant clinical studies in paediatric resuscitation, making comparator data scarce.

Prescribing error rates in the (non-simulated) emergency setting have been reported to be between 10.1% and 16%[24 25]

of all orders; our study reports a lower rate of 5%, although this difference may be at least partly due to different error definitions. Our study reinforces similar findings in another recent analysis[26] suggesting that preparation and administration errors may be more common, but is the first to highlight the extent to which these errors go undetected. Other simulated studies have reported error rates for the administration of intravenous bolus medication of between 15.5% and 26.5%[4 5 27]; in our study, it was 31%. The referenced studies, however, reported only on dose errors, and not any other medication error types. Only seven out of the 24 medication errors we observed for bolus doses were dose errors. Medications given by intermittent infusion were the most error-prone in our study. There are no studies that investigate emergency administration of intermittent infusions in sufficient detail to provide a basis against which to compare this finding.

Medications given by continuous infusion are potentially the most complicated in paediatric emergencies. In addition to the preparation steps for intermittent infusions, staff generally have to convert infusion rates in milligrams or micrograms per kilogram per minute to infusion rates of millilitres per hour. A recent trial of a digital application reported errors in 70% of continuous vasopressor infusions[6] in the control arm. However, despite the increased cognitive demand, we observed the lowest incidence of medication error for continuous infusions, at 18%. Administration of continuous infusions in our hospital seemed to be relatively well supported by an online/paper tool.[28]

## Limitations

Although considerable efforts were made to replicate the paediatric emergency department environment in a state-of-the-art facility, the simulation environment may not reflect the clinical environment during a genuine emergency. Participants were not blinded to the purpose of the study, and therefore this investigation is potentially subject to preparation bias.

This investigation was conducted at a single site and all three of the nurse video assessors were from the same academic unit in the study hospital. It is thus possible that, despite best efforts at standardisation, the results may be skewed to reflect the expectations of medication practice at the study site.

Although this is one of the largest simulations studies in paediatric emergency medicine investigating medication errors, the sample size made it impossible to make probabilistic assessments of the relationship between step discrepancies and medication errors. Furthermore, the discrepancy assessment was made by a single nurse assessor. Save for the extract on which inter-rater reliability was tested, we did not have the resources to evaluate 884 discrepancies by more than one clinician, reaching consensus on each.

Finally, we did not investigate the potential role of clinical pharmacists in the resuscitation setting. There is convincing evidence that the presence of clinical pharmacists reduces medication errors in this setting,[29–31] and the emergency department at the study site does, at times, benefit from the assistance and expertise of clinical pharmacists. However, their presence is not routine during resuscitations.

## Implications for research and practice

This study highlights the need for research to optimise clinicians' use of electronic resources containing medication preparation and administration information. We were not able to pinpoint the precise steps at which the current electronic intravenous medications guidance system in the study hospital proved vulnerable to misinterpretation. Research to further understand the steps that need attention may serve as a useful basis from which to refine, and if needed, redesign such systems.

This study reaffirms that performing complex arithmetic in high-stress clinical environments is a considerable contributor to medication error.[32] With the purpose of addressing medication safety in paediatric resuscitation, the literature has been dominated by studies looking at 'resuscitation aids', most commonly length-based tapes.[18 33 34] These aids couple weight estimation with a suggested dose for a limited number of medications, but usually do not provide comprehensive preparation and administration support. It is not likely that length-based tapes would have decreased the rate of medication error in this study. Further clinical research is required to determine the effectiveness of new digital tools that do support preparation and administration, such as those that have shown promising results in simulated studies.[6 35]

Human factors methods have been used in other high-risk industries to define system vulnerabilities for building safer systems. By using quantitative HRA, this study provides evidence for the prioritisation of research efforts directed towards new interventions to address the most important system weaknesses.

In terms of implications for practice, one of the most unexpected findings in this investigation was the uncovering of 'purely mechanical' task discrepancies resulting in medication errors. During drug preparation, clinicians were observed drawing up incorrect volumes of medications or diluents even though all calculations were correct. This suggests that efforts seeking to address medication safety in cognitively demanding environments using clinical education strategies or contemporary technologies must do so without disregarding the seemingly 'simplest' aspects of drug preparation. The reliability of information exchanges between healthcare professionals similarly needs improvement. Verbal medication orders in particular are inconsistent and error-ridden. Particular attention should be paid to medication orders given verbally in the emergency setting, using approaches such as the recipient verbally confirming the medication and dose being prepared. More importantly however, there is an urgent need for research to explore how to bring greater effectiveness to checking and double-checking more broadly. These are steps intended to defend patients from error, but which are too often ineffective.

## CONCLUSIONS

Overall, we identified errors in 29% of all simulated medication administrations, only two of which were detected by participants, with 40% of these likely to result in moderate or severe harm. HRA revealed a number of error-prone steps, many of which occurred during preparation and administration of correctly ordered medications. The task most likely to result in erroneous medication administration was ineffective retrieval of correct medication preparation and administration instructions from intravenous medication guidance.

This study has highlighted an urgent need to optimise existing systems and to commission new approaches to increase the reliability of human interactions with the emergency medication administration process.

**Author affiliations**
[1]Department of Surgery and Cancer, Division of Surgery, Imperial College London, London, UK
[2]NIHR-Imperial Patient Safety Translational Research Centre, Imperial College London, London, UK
[3]Helix Centre for Design in Healthcare, Imperial College London, London, UK
[4]Centre for Mathematics of Precision Healthcare, Imperial College London, London, UK
[5]School of Pharmacy, University College London, London, UK
[6]Department of Paediatric Intensive Care, Division of Women and Children's Services, Imperial College Healthcare NHS Trust, London, UK
[7]Department of Emergency Medicine, Division of Medicine, Imperial College London, London, UK

# Open access

**Acknowledgements**  We would like to thank all clinical staff who participated in the simulations. It is our hope that the enthusiasm and extraordinary open-mindedness by our dedicated and deeply competent colleagues demonstrated in making this study possible will lead to safer systems and innovations in paediatric medication safety. This study was funded by the NIHR-Imperial Patient Safety Translational Research Centre, with infrastructure support from the NIHR Imperial Biomedical Research Centre. Support was also received from the Resuscitation Council (UK).

**Contributors**  NA, CF, JC and RS designed the study. RS facilitated the simulations, with NA and CF observing and coordinating. AD provided funding. CF analysed the video footage. NA and JC were responsible for the data analysis which was reviewed by PP. BF and IM advised on error definitions and analysis. NA drafted the study report, which was reviewed by all authors. All authors read and approved the final draft.

**Funding**  NA received funding from the Resuscitation Council (UK) which partially funded elements of the simulations and data extraction, with the majority of the funding received from the National Institute for Health Research (NIHR) Imperial Patient Safety Translational Research Centre. Infrastructure support was provided by the NIHR Imperial Biomedical Research Centre (BRC). BF is also supported by the NIHR Health Protection Research Unit in Healthcare Associated Infections and Antimicrobial Resistance at Imperial College London, in partnership with Public Health England (PHE). The views expressed are those of the authors and not necessarily those of the NHS, PHE, the NIHR or the Department of Health and Social Care.

**Competing interests**  The Helix Centre at Imperial College London is leading an effort in collaboration with the British National Formulary developing digital tools in an attempt to improve paediatric medication safety. NA has written two patents describing syringe labelling techniques in medication safety. BF supervises a PhD student part funded by a supplier of hospital electronic health record systems and has received funding from Pfizer for organising and chairing two symposia on medication safety. We confirm that we have given due consideration to the protection of intellectual property associated with this work and that there are no impediments to publication, including the timing of publication, with respect to intellectual property. In so doing we confirm that we have followed the regulations of our institution concerning intellectual property.

**Patient consent for publication**  Not required.

**Ethics approval**  The Health Research Authority approved this study.

**Provenance and peer review**  Not commissioned; externally peer reviewed.

**Data availability statement**  Data are available upon reasonable request.

**ORCID iDs**
Nicholas Appelbaum https://orcid.org/0000-0002-9458-6828
Jonathan Clarke https://orcid.org/0000-0003-1495-7746
Calandra Feather http://orcid.org/0000-0003-1322-6589

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
