## [Reviewer comments · BMJ Open]

ARTICLE DETAILS

TITLE (PROVISIONAL)	Medication errors during simulated paediatric resuscitations: a prospective, observational human reliability analysis
AUTHORS	Appelbaum, Nicholas; Clarke, Jonathan; Feather, Calandra; Franklin, Bryony; Sinha, Ruchi; Pratt, Phillip; Maconochie, Ian; Darzi, Ara;

VERSION 1 – REVIEW

REVIEWER	Ronald. S. Litman The Children's Hospital of Philadelphia, USA
REVIEW RETURNED	17-Jul-2019

GENERAL COMMENTS	This manuscript represents an analysis of medication errors during two different types of simulated pediatric resuscitation events. Overall, it's a well-performed and well-written study that reveals an astonishing, yet believably high incidence of errors. The major limitation, as the authors acknowledge, is the bias among the study participants, and this reviewer would like the authors to at least make a statement as to why this is so. For example, in my institution, blinding of a simulation team is possible prospectively, as long as the purpose of the blinding was revealed and discussed in the post-event debriefing sessions. Minor critiques include: - The pdf graphs of Figures 2 and 3 were not readable due to poor resolution so I could not assess properly.- Page 26: lines 2-22: It would be nice (although not applicable to the research methodology) if the authors could reference the "standard paediatric guidelines" – I only mention that because some of their treatments differ slightly from those here in the U.S.- Page 6, line 59: what do the authors mean when they speak of "Face validity" here? Does that mean the experts agreed on the treatment protocol? Or the ability of the team to adhere to it?- Page 9, lines 31-44: was there only one researcher who assessed these causality categories? In most studies that I know of in this area of research, there is more than one assessor, that come to agreement by consensus.- The expert panel was mentioned three times in the paper – I assume it was the same panel throughout?- An obvious omission is the lack of discussion of specific solutions. Perhaps this is best left for an accompanying editorial.
--

REVIEWER	Dr Magdalena Raban Australian Institute of Health Innovation, Macquarie University
REVIEW RETURNED	30-Aug-2019

GENERAL COMMENTS

Thank you for the opportunity to review this manuscript reporting an analysis of medication errors in simulated emergency scenarios. Very little is known about medication errors in emergency scenarios, so this manuscript makes a valuable contribution. I have made some suggestions mostly to improve the clarity of the methods and results.

Major comments

The definition of a discrepancy in the manuscript is not clear to me. Often the terms error and discrepancy are used interchangeably in the literature on medication errors, and thus I think it is important for the authors to be clear on the difference here. The first sentence under "Outcome measures" in the Methods attempts to define medication error vs a discrepancy. However, what is an 'observed deviation at the level of the individual task'? Deviation from what? Is it the deviation from the expected steps in the medication process? Perhaps some examples here would help. Table 4 helps somewhat, but I still found it hard to interpret the various discrepancies listed. e.g. how can there be a discrepancy in the step 'dose from memory'? Or 'program infusion pump'? Perhaps more detailed definitions can be added to the appendix if appropriate.

Results, 'Errors and discrepancies during medication preparation', second para: similar to above, it seems to me that the last two examples relating to the incorrect amount of diluent or drug in a syringe are an error, rather than discrepancy. But this may be because I am not clear on the discrepancy definition.

The definition of a 'large magnitude error' was also unclear to me.

The authors refer to an assessment of error severity using Dean and Barber's tool. However, using the term 'error severity' makes is ambiguous as to whether potential or actual harm was assessed. Despite this being a simulated environment, I think it would be clearer to refer to this as an assessment of the level of potential harm from the errors.

IRR methods: the authors state that methods used for continuous and categorical variables. It would be useful to provide the specific variables in brackets here to be clear. Also, how many variables were assessed of each type?

Minor comments

Abstract: consider revising the term 'individual process steps' as it lacks clarity. In the results, briefly define 'large magnitude' errors.

Background, page 5, lines 4-7: is it possible to add a reference for the statement that medication errors are under-detected and under-reported?

I think it would be clearer to say "route of administration" rather than "method of administration".

In the Methods (page 8) the authors introduce the abbreviation tDRD. However, this is used in this section of the main manuscript (and only twice), so perhaps is not needed? Especially since it is similar to other acronyms used throughout (e.g. DRDR, DRDRate).

Table 3: consider adding the medication stage during which the error occurred i.e. prescribing. Also, for the two errors referring to a % of

	median time, consider adding “of” after the %. Discussion, third paragraph: the authors compare their results to other simulation studies. Were these also conducted with emergency scenarios? Discussion: I did not see a limitation section in the Discussion. Could discuss the fact this was a simulation study (even though a study in the real-world would have low feasibility). I could not review Fig 3 as the resolution was too low to zoom in.
--	--

REVIEWER	Rioufol, Catherine Hospices Civils de Lyon, France
REVIEW RETURNED	01-Sep-2019

GENERAL COMMENTS	Very important and interesting paper to be published as soon as possible Very good level of methods regarding on bibliography and panels of experts My single regret is that the steps of medication errors and discrepancies do not include pharmaceutical dispensation that is yet a step at risks
---

VERSION 1 – AUTHOR RESPONSE

Reviewer 1:	Ronald. S. Litman Institution and Country: The Children’s Hospital of Philadelphia, USA
	Thank you for your positive feedback regarding the manuscript.
The major limitation, as the authors acknowledge, is the bias among the study participants, and this reviewer would like the authors to at least make a statement as to why this is so. For example, in my institution, blinding of a simulation team is possible prospectively, as long as the purpose of the blinding was revealed and discussed in the post-event debriefing sessions.	As you have helpfully pointed out, this was inadequately addressed in the manuscript. In our institution it is not the norm to blind simulation participants for studies such as this. Given the fact that simulation sessions were held over many weeks, the team considered that it was inevitable that teams would discuss the study between themselves. Preparation bias, were it to have occurred would more likely have decreased the rate of observed error. The paragraphs addressing bias in the manuscript now read “Although considerable efforts were made to replicate the paediatric ED environment in a state-of-the-art facility, the simulation environment may not reflect the clinical environment during a genuine emergency. Participants were not blinded to the purpose of the study, and therefore this investigation is potentially subject to preparation bias. This investigation was conducted at a single site and all three of the nurse video assessors were from the same academic unit in the study hospital. It is thus possible that,

	despite best efforts at standardisation, the results may be skewed to reflect the expectations of medication practice at the study site.”
The pdf graphs of Figures 2 and 3 were not readable due to poor resolution so I could not assess properly.	We apologise for this unfortunate technical error. High quality PDF's have now been uploaded.
Page 26: lines 2-22: It would be nice (although not applicable to the research methodology) if the authors could reference the “standard paediatric guidelines” – I only mention that because some of their treatments differ slightly from those here in the U.S.	A reference for the British National Formulary has been added to the manuscript. Local guidelines are typically locally produced PDF documents based on the relevant national guidelines or other evidence. It is unfortunately not possible to reference or add the local guideline document as an appendix as they are not in the public domain.
Page 6, line 59: what do the authors mean when they speak of “Face validity” here? Does that mean the experts agreed on the treatment protocol? Or the ability of the team to adhere to it?	Thank you for suggesting this clarification. Detail has been added and the manuscript now reads “Face validity was established by an independent expert panel of six, with representation from each of these professional groups, including two lead paediatric clinical nurse educators. It was deemed that the two scenarios were both similarly demanding and clinically sound, with treatment recommendations corresponding closely with Resuscitation Council (UK) and Royal College of Paediatrics and Child Health teaching cases.”
Page 9, lines 31-44: was there only one researcher who assessed these causality categories? In most studies that I know of in this area of research, there is more than one assessor, that come to agreement by consensus.	There was only one researcher who made the causality assessment across the full dataset, other than the single simulations (1/15) which was assessed by an additional assessor and tested for interrater reliability. Unfortunately, the grant that funded this study did not provide the resources to evaluate 884 discrepancies by reaching consensus on each.
The expert panel was mentioned three times in the paper – I assume it was the same panel throughout?	This was the same expert panel. This has been clarified in the manuscript by adding the word ‘same’.
An obvious omission is the lack of discussion of specific solutions. Perhaps this is best left for an accompanying editorial.	The solutions to this problem are an area of intense research for our institute and our group more specifically. Unfortunately, this manuscript already runs long on word count. In order to do this important topic justice, we feel that a separate publication is required.
Reviewer 2:	Dr Magdalena Raban Institution and Country: Australian Institute of Health Innovation, Macquarie University
The definition of a discrepancy in the manuscript is not clear to me. Often the terms error and discrepancy are used interchangeably in the literature on medication errors, and thus I think it is important for the authors to be clear on the	We thank the reviewer for this extremely valid observation. The medication safety literature is plagued with inconsistency of terminology in this respect and it's our hope that our paper is as clear as possible on this point.

difference here. The first sentence under “Outcome measures” in the Methods attempts to define medication error vs a discrepancy. However, what is an ‘observed deviation at the level of the individual task’? Deviation from what? Is it the deviation from the expected steps in the medication process? Perhaps some examples here would help. Table 4 helps somewhat, but I still found it hard to interpret the various discrepancies listed. e.g. how can there be a discrepancy in the step ‘dose from memory’? Or ‘program infusion pump’? Perhaps more detailed definitions can be added to the appendix if appropriate.	We have substantially edited the relevant section in the manuscript, added a reference to a paper which has used a similar approach (BMJ Quality and Safety) and added examples to help distinguish between discrepancies and medication errors. The relevant section now reads: “We reserved the term ‘medication error’ to describe an overall error with respect to a particular drug administration as a whole, after having been administered to a patient, and the term ‘discrepancy’ to refer to observed deviations from expected local practice at the level of the individual task. This approach has been successfully used by other medication safety researchers.¹⁷ Variation in clinician practice and human adaption to the complex process of medication ordering, preparation and administration typically results in many minor task discrepancies that may not individually, or even in combination, result in a medication error or patient harm.¹⁷ Other discrepancies almost always result in medication errors. For example, a prescribing or pump-programming discrepancy is highly likely to result in a medication error. To identify the most important task discrepancies, we assessed all observed discrepancies to establish the extent to which they may or may not have contributed to any resultant medication errors.”
Results, ‘Errors and discrepancies during medication preparation’, second para: similar to above, it seems to me that the last two examples relating to the incorrect amount of diluent or drug in a syringe are an error, rather than discrepancy. But this may be because I am not clear on the discrepancy definition.	Hopefully the clarification above will have addressed this concern.
The definition of a ‘large magnitude error’ was also unclear to me.	Thank you for pointing out this omission of an important definition. We have amended the relevant section in ‘outcome measures’ to read: “Where there was a greater than 25% discrepancy in the DRDR or DRDRate, the errors were considered as ‘large magnitude’.” Further clarification has also been added as a reminder, in parentheses, in the results section.
The authors refer to an assessment of error severity using Dean and Barber’s tool. However, using the term ‘error severity’ makes is ambiguous as to whether potential or actual harm was assessed. Despite this being a simulated environment, I think it would be clearer to refer to this as an assessment of the level of potential	Thank you for this suggestion. This simulated study was, of course, only able to assess potential harm. We have made this more clear by adding the word ‘potential’ to the manuscript, which now reads: “There are few validated tools that can be used

harm from the errors.	to assess the potential severity of medication errors without knowledge of actual patient outcomes and that are thus usable in simulated studies¹⁷. One of these tools is that of Dean and Barber...”
RR methods: the authors state that methods used for continuous and categorical variables. It would be useful to provide the specific variables in brackets here to be clear. Also, how many variables were assessed of each type?	We have added the number of variables and examples to the section describing the inter observer reliability section. The manuscript now reads: “One of the 15 simulations was re-analysed by an additional independent nurse not involved in the simulations. Spearman’s rank correlation coefficient was calculated for 38 continuous variables (e.g. doses, elapsed times) and Cohen’s Kappa for 20 categorical variables (e.g. reconstitution fluid, labelling assessment). The dataset concerning the medication errors and timing parameters was treated separately from the dataset containing the parameters for the HRA (the task discrepancies).”
Abstract: consider revising the term ‘individual process steps’ as it lacks clarity. In the results, briefly define ‘large magnitude’ errors.	Thank you for this suggestion. We have amended the abstract in-line with both of these points. A parenthesized definition for large magnitude has been added as “(>25% discrepant)”. The objectives section now reads: “To describe the incidence, nature and severity of medication errors in simulated paediatric resuscitations, and to employ human reliability analysis to understand the contribution of discrepancies in individual process steps to the occurrence of these errors.”
Background, page 5, lines 4-7: is it possible to add a reference for the statement that medication errors are under-detected and under-reported?	A reference has been added to the manuscript.
I think it would be clearer to say “route of administration” rather than “method of administration”.	Thank you for raising this concern which we have hopefully addressed adequately. Route of administration and method of administration are not the same thing and the terms are not interchangeable. Routes of administration are We have now made this clearer in the methods section by listing them separately as “...route of administration, or method of administration...”.
In the Methods (page 8) the authors introduce the abbreviation tDRD. However, this is used in this section of the main manuscript (and only twice), so perhaps is not needed? Especially since it is similar to other acronyms used throughout (e.g. DRDR, DRDRate).	Thank you for this suggestion. tDRD has been removed from the manuscript with the relevant section now reading: “The time to be ready for deliver was considered ‘prolonged’ when a particular team took more than double the median time for that specific drug across the

	entire study without clinical cause for the delay as determined by the nurse assessor. For example, if a medication administration was interrupted to reassess the patient clinically or to administer another medication as a priority, a prolonged time would be excluded as an error on clinical grounds.”
Table 3: consider adding the medication stage during which the error occurred i.e. prescribing. Also, for the two errors referring to a % of median time, consider adding “of” after the %.	Thank you for this suggestion too which dovetails well with the overall narrative of the paper. We have added a ‘Stage’ column to Table 3. ‘of’ has been added before ‘%’ in the two relevant places.
Discussion, third paragraph: the authors compare their results to other simulation studies. Were these also conducted with emergency scenarios?	The studies reporting on rates of error during medication ordering were from non-simulated studies, but those investigating errors during preparation and administration were conducted in the simulated setting. ‘Non-simulated’ has been added in parentheses to add clarity to the relevant paragraph which now reads: “Prescribing error rates in the (non-simulated) emergency setting have been reported to be between 10.1% and 16% ^{24,25} of all orders; our study reports a lower rate of 5%, although this difference may be at least partly due to different error definitions. Our study reinforces similar findings in another recent analysis ²⁶ suggesting that preparation and administration errors may be more common, but is the first to highlight the extent to which these errors go undetected. Other simulated studies have reported error rates for the administration of intravenous bolus medication of between 15.5% and 26.5% ^{4,5,27} ; in our study, it was 31%.”
Discussion: I did not see a limitation section in the Discussion. Could discuss the fact this was a simulation study (even though a study in the real-world would have low feasibility).	A limitation section has now been added which includes comment on the shortcomings of simulated research.
I could not review Fig 3 as the resolution was too low to zoom in.	We apologise again for this technical error. High quality figures accompany this resubmission.
Reviewer 3:	Catherine Rioufol Institution and Country: Hospices Civils de Lyon, France
Very important and interesting paper to be published as soon as possible Very good level of methods regarding on bibliography and panels of experts	Thank you for these positive comments.
My single regret is that the steps of medication errors and discrepancies do not include pharmaceutical dispensation that is yet a step at risks	Thank you for this suggestion. However, dispensing in the pharmacy department was out of scope for the present study as this would need a very different study design.

VERSION 2 – REVIEW

REVIEWER	Ron Litman Children's Hospital of Philadelphia, Institute for Safe Medication Practices
REVIEW RETURNED	05-Oct-2019

GENERAL COMMENTS	Thank you for a nice revision.
--------------------------------

REVIEWER	Dr Magdalena Raban Centre for Health Systems and Safety Research, Australian Institute of Health Innovation, Macquarie University
REVIEW RETURNED	09-Oct-2019

GENERAL COMMENTS	The changes made to the manuscript have improved an already strong submission.
--